# Nutritional Status Assessment of Newborns: Comparison of the CAN Score (Metcoff Methodology), Growth Curves, Anthropometry, and Plicometry

**DOI:** 10.3390/nu17101642

**Published:** 2025-05-12

**Authors:** Maria L. Felix, Carmen Basantes, Susana Nicola, Susana Hidalgo, Patricia Guevara-Ramírez, Santiago Cadena-Ullauri, Ana Karina Zambrano

**Affiliations:** 1Facultad de Ciencias de la Salud Eugenio Espejo, Universidad UTE, Quito 170129, Pichincha, Ecuador; 2Centro de Investigación en Salud Pública y Epidemiología Clínica (CISPEC), Facultad de Ciencias de la Salud Eugenio Espejo, Universidad UTE, Quito 170129, Pichincha, Ecuador; 3Centro de Investigación Genética y Genómica, Facultad de Ciencias de la Salud Eugenio Espejo, Universidad UTE, Quito 170129, Pichincha, Ecuador

**Keywords:** healthcare, malnutrition, vulnerable groups, growth standards, preventive care

## Abstract

Fetal malnutrition, characterized by inadequate fat and muscle accretion during intrauterine development, has been linked to adverse outcomes, ranging from neonatal complications to long-term developmental and metabolic disorders. Traditionally, growth curves and birth weight have guided the assessment of newborns’ nutritional status; however, these measures often do not accurately reflect changes in body composition. This review compares several evaluation methods—CAN score (Metcoff methodology), body mass index (BMI), Ponderal Index (PI), McLaren Index, mid–upper arm circumference (MUAC), and plicometry—to provide suggestions on selecting the most appropriate approach, depending on the healthcare setting and population needs. Findings from multiple international studies indicate that the CAN score and BMI are among the most accurate tools, offering better sensitivity and specificity than traditional anthropometric indicators. The CAN score, based on a clinical observation of fat deposits, skin texture, and muscle tone, has been widely used in Latin America and remains a practical and cost-effective option. Nonetheless, recent research suggests that BMI, mainly when used alongside the PI, may outperform the CAN score in certain contexts. Considering the complexity of fetal nutritional assessments, integrating multiple methods enhances the diagnostic accuracy. Early identification of malnourished newborns is essential for timely intervention and improved long-term outcomes. Standardizing these diagnostic tools globally could advance efforts to reduce neonatal morbidity and mortality by 2030.

## 1. Introduction

Fetal and neonatal malnutrition remains a major public health concern globally, especially affecting low- and middle-income countries. Fetal malnutrition has been associated with higher risks of miscarriage, intrauterine growth restriction, anemia, and low birth weight [1,2]. In Latin America, inequalities in access to healthcare systems further exacerbate these situations, complicating the accurate assessment of newborns’ nutritional status. Neonates are particularly vulnerable to malnutrition, which can negatively impact both physical and neurocognitive development, increasing the risk of neurodevelopmental disorders, learning difficulties, and delayed motor milestones [3].

Clinical signs of neonatal malnutrition include fragile and sparse hair, a triangular facial appearance, loose and wrinkled skin around the neck, thin arms with accordion-like wrinkles, a lax-skinned abdomen, thin legs with accordion-like wrinkles, and fat loss over the back and flaccidity in the buttocks. These signs often coexist with wasting and growth retardation, conditions associated with higher rates of infant morbidity and mortality [4,5]. Over time, fetal malnutrition may also contribute to the onset of chronic diseases such as hypertension, type 2 diabetes, and cardiovascular diseases—a phenomenon known as “fetal programming” [6,7].

The primary causes of fetal malnutrition are inadequate maternal nutrition and placental insufficiency. These factors hinder the deposition of subcutaneous fat and muscle mass deposits during gestation [8,9], often resulting in excess skin on the neck, limbs, and buttocks and, in severe cases, an aged face of an emaciated infant [10].

Fetal malnutrition is estimated to affect 2% to 10% of all live births worldwide [11,12]. In 2022, approximately 149 million children under the age of five were affected by stunting, while approximately 45 million experienced wasting, reflecting the inadequate nutritional status in this age group [13,14]. Additionally, 19.1% of women of childbearing age were found to be anemic—a condition that can affect fetal growth and development [14]. In Latin America, around seven million children suffer from malnutrition. Countries such as Guatemala, Guyana, and Haiti report the highest prevalence, with over 10% of children affected. In contrast, Argentina, Brazil, Chile, and Jamaica have reported rates below 2.5% [13].

According to the Economic Commission for Latin America (CEPAL, by its Spanish acronym), the incidence of neonatal malnutrition has decreased; however, over 10% of newborns are still born with a low birth weight, and 5% experience intrauterine growth restriction [15]. The data on the incidence of neonatal malnutrition are limited, but estimates indicate that 14.6% of infants are born with a low birth weight [16]. These data highlight the need to implement more precise nutritional assessment methodologies tailored to the specific characteristics of neonates.

Among the available tools, neonatal anthropometry, growth curves, plicometry, and the Clinical Assessment of Nutritional Status (CAN score) methodology are the most used [15,17,18]. However, concerns about their sensitivity and specificity remain—especially in low- and middle-income countries—where disparities in healthcare access and variations in clinical practices may impact the early identification and effective intervention of malnutrition cases [19,20]. This highlights the importance of validating these methodologies to ensure their applicability in populations at high risk of malnutrition.

This article describes and addresses the gap in the comparative analysis among existing tools used for assessing the neonatal nutritional status. Furthermore, despite the importance of this assessment, the existing literature remains limited and often does not include crucial context-specific evaluations. The purpose of the present review is to perform a comprehensive review of the current neonatal nutrition assessment methodologies by analyzing the strengths and limitations of each approach and provide suggestions on selecting the most appropriate approach, depending on the healthcare setting and population needs.

## 2. Methodology

A comprehensive literature review was conducted using databases such as PubMed, SciELO, Redalyc, and Google Scholar. The search included keywords and MeSH terms such as neonatal malnutrition, neonatal nutritional assessment, Metcoff methodology or CAN score, growth curves, anthropometry, and plicometry.

The evidence reviewed in this article comes mainly from observational studies (prospective, retrospective, cross-sectional, and descriptive designs) and secondary analyses such as meta-analyses and systematic reviews. It is important to clarify that no clinical trials were included, as the evaluation of diagnostic tools for fetal malnutrition is typically conducted through observational research rather than interventional designs.

## 3. Description of Methodologies for the Evaluation of the Neonatal Nutritional Status

### 3.1. Anthropometry

For decades, anthropometry has been the most widely used method for assessing the neonatal nutritional status due to its accessibility and clinical applicability. This approach is based on measuring various body dimensions and comparing them to national and international standards, allowing for the identification of potential deviations in the growth of newborns. The most used measurements at birth include weight, length, and head, arm, thigh, and chest circumferences. These parameters have been used to construct growth curves for different gestational ages, facilitating fetal and neonatal growth monitoring [21]. In this context, birth weight determination has enabled the establishment of widely used classifications in clinical practice, differentiating neonates into categories such as low birth weight, very low birth weight, extremely low birth weight, adequate weight, and high birth weight [22].

The relationship between weight and gestational age has been a fundamental criterion for neonatal nutritional assessment. Based on this parameter, newborns can be classified as appropriate for gestational age (AGA), between the 10th and 90th percentiles; large for gestational age (LGA), above the 90th percentile; and small for gestational age (SGA), below the 10th percentile [23]. Nevertheless, although birth weight is a key indicator of nutritional status, its isolated use does not accurately determine the neonate’s body composition [10]. In recent years, the addition of complementary methods to assess neonatal nutrition has been encouraged, as relying on a single anthropometric measurement is inadequate for accurately determining a newborn’s nutritional status [24].

#### 3.1.1. Mid–Upper Arm Circumference (MUAC)

The mid–upper arm circumference is a useful anthropometric indicator that provides both the muscle mass composition and fat reserves in the arm, offering a practical measure for overall body adiposity. Its sensitivity to rapid shifts in protein and fat stores makes it particularly valuable for monitoring the nutritional status of neonates. The upper limb is the preferred site for measurement due to its lower susceptibility to changes in body fluids compared to other body regions. This is particularly relevant in neonates with edema, where fluid fluctuations can affect the accuracy of other anthropometric measurements. Notably, specific growth standards like Rolland-Cachera and Sasanow curves have been developed to contrast the values with the corresponding gestational ages [25,26,27].

#### 3.1.2. Thigh Circumference (TC)

TC is not a commonly assessed metric in preterm newborns [28]. Ashton et al. established significant positive correlations between the TC, MUAC, weight, length, and head circumference (HC) [29]. The utility of MUAC and TC in preterm newborns warrants further investigation because it is not clear how these measures are related to body composition. Term and preterm infant reference curves for MUAC and TC are available, including Merlob and Rached-Sosa curves [27,28,30,31].

Kanawati–McLaren Index or Mid–Upper Arm Circumference/Head Circumference (MUAC/HC) Ratio

This ratio is an anthropometric tool to assess children’s nutritional status. It is calculated by dividing the MUAC measurement, expressed in centimeters with millimeter precision, by the HC measurement with the same level of precision [32]. This ratio helps estimate the body correlation, combining a variable highly influenced by nutritional intake, the MUAC, with another one that remains more stable despite variations, the HC [25]. This balance between sensitivity and stability makes the MUAC/HC index particularly useful in identifying varying degrees of malnutrition in children older than 3 months. Standardized growth curves, such as Rached-Sosa curve have been designed for this approach; however, its application is limited due to a reduced sensitivity and particular cases, such as microcephaly [25,31,33].

### 3.2. CAN Score or Metcoff Methodology

The method was developed by Jack Metcoff in 1994, building upon the earlier observations of McLean, Usher, and Scott, whose findings were published between 1966 and 1970. This methodology has become a crucial tool for assessing the neonatal nutritional status. It is based on identifying nine signs of malnutrition in the neonate’s body, which are classified on a scale ranging from four (no evidence of malnutrition) to one (clear signs of intrauterine malnutrition). According to Metcoff’s guidelines, a score of 24 or higher indicates a preserved nutritional status, while a score below 24 indicates the presence of malnutrition (Figure 1) [34].

This method has demonstrated effectiveness in assessing body composition, allowing a more accurate diagnosis of neonatal malnutrition [34]. It has been widely used in several countries, particularly to differentiate between SGA newborns without malnutrition and those with malnutrition [35]. Although this approach is mainly used for term neonates, some studies have demonstrated its usefulness in preterm infants, albeit with slightly reduced sensitivity [36]. In fact, in several studies, this method has been considered the gold standard for evaluating neonatal malnutrition, enabling comparisons of the predictive value of other instruments [10].

In neonates with adequate or high weight for gestational age, the application of this method has demonstrated the ability to detect fetal malnutrition in 15.5% to 12.3% of neonates with AGA [10,37] and in 0.28% to 2.2% of neonates with high weight for gestational age [38,39].

The CAN score presents several advantages that support its utility in the nutritional assessment of neonates. Notably, it is simple to apply, requires no specialized equipment, and serves as a cost-effective and accessible tool for use in a wide range of clinical settings. Additionally, its simplicity and speed allow for continuous monitoring of nutritional changes without the need for complex calculations. The CAN score can also be applied regardless of weight for gestational age, expanding its usefulness in assessing neonates with different growth characteristics [10,35].

However, a key limitation of the CAN score is its dependency on the operator, as its accuracy is related to the experience and training of the professional. Studies have shown that there can be variability in the scores assigned by different evaluators, which may affect the score’s reproducibility and consistency [40].

### 3.3. Plicometry

Plicometry is a methodology for assessing the nutritional status and body, offering valuable insights into evaluating individual growth [41]. This method relies on measuring the thickness of skinfolds at various body sites, allowing for an estimation of the body composition, particularly the body fat percentage [41]. Its non-invasive nature, safety, and ease of access and interpretation make it a useful clinical tool for determining the neonatal nutritional status [42,43]. In the case where the subcutaneous fat tissue is reduced, it is associated with a delay in intrauterine growth, whereas fat tissue of the abdominal wall is not correlated, indicating that various factors determine the place for fat accumulation during intrauterine growth. In neonates, plicometry typically includes measurements of four or five skinfolds, with the most assessed being the subscapular, triceps, biceps, and suprailiac folds, while the least assessed is the quadriceps. The suprailiac fold is particularly useful to monitor fat mass in full-term and premature newborns. However, these measurements may not be appropriate for extremely premature neonates [44]. Despite its advantages, plicometry also has limitations, including being time-consuming, in comparison with the CAN score, a factor that should be considered in areas with limited healthcare personnel. Additionally, its accuracy depends on the evaluator’s skill, highlighting the need for trained personnel to ensure greater reliability in the measurements [40].

Standards like the Rodríguez curve have been established to compare the values obtained from the sum of folds, leading to the establishment of cut-off values for the 10th and 90th percentiles: 5.6 mm to 11.2 mm for girls and 6.7 mm to 11.9 mm for boys from 33 to 41 weeks of gestational age [45]. Moreover, plicometry allows the assessment of the ratio of central subcutaneous tissue to total folds (subscapular + suprailiac/total subcutaneous tissue (sum of folds) × 100) for each sex and for neonates from 33 to 41 weeks. Although these tables establish cut-off points, the study has limitations due to the small number of patients younger than 35 weeks who were included [45].

A cross-sectional anthropometric study conducted on 10,226 Chinese neonates from 28 to 42 weeks gestational age correlated sex anthropometric variables (PI, BMI, TC/HC, weight/HC, weight/height, and Kanawaki–McLaren index) and established that the weight/height ratio had the highest correlation with the fold sum value [46].

Furthermore, a strong correlation between body fat determined by skinfold thickness (SFT) and dual energy x-ray absorptiometry (DXA) has been reported in infants at birth [47]. For instance, Daly-Wolfe et al. observed a high correlation between suprailiac skinfold and body fat measurements determined by air displacement plethysmography (ADP) [48]. Nevertheless, these tools are rarely available for the routine evaluation of infants [49].

### 3.4. Ponderal Index (PI)

The *PI*, proposed by Rohrer in 1960, is a tool used to indirectly estimate neonatal adiposity and differentiate between symmetrical and asymmetrical SGA neonates [50]. It is calculated using the following formula:$$PI=\frac{Weight(g)}{{Height}^{3}({\mathrm{cm}}^{3})}\times 100$$

Growth curves for IP/gestational age, such as Rached-Sosa, allow an increased precision in SGA newborns’ diagnosis and determine whether they are symmetrical (low weight and length) or asymmetrical (normal weight and length) [31,51].

The generally accepted cut-off point for defining fetal malnutrition is a *PI* of less than 2.2 g/cm^3^. In the classification of fetal growth disorders, newborns with a *PI* between 2.32 and 2.85 are considered symmetrical, while those with a *PI* above the 90th percentile are classified as overweight. Neonates below the 10th percentile with a *PI* between 2.15 and 2.31 are classified as wasted. Additionally, a *PI* below the third percentile for gestational age and sex is indicative of the SGA status [52]. Studies have reported a diagnostic sensitivity of 65% and a specificity of 93% for detecting malnutrition [53].

### 3.5. Body Mass Index (BMI)

Neonatal *BMI* relates weight to height, allowing for a more accurate assessment of the nutritional status at any age. Its application in the neonatal population has been the subject of multiple studies, including research by Brock et al. (2008), who established specific reference values for neonates and expressed them in sex-differentiated percentile curves [24]. Notably, a growth curve has been developed to assess Native American populations [31].$$BMI=\frac{Weight(\mathrm{kg})}{{\left[Height(m)\right]}^{2}}$$

Table 1 represents a comparison between tools and indices used to diagnose the nutritional status of newborns.

### 3.6. Growth Curves

Monitoring neonatal growth is essential for assessing the nutritional status. Growth curves serve as fundamental tools in the clinical follow-up of neonates, particularly in neonatal intensive care units [54]. However, there is no international consensus on the optimal methodology for evaluating neonatal growth, leading to the implementation of different approaches [55,56].

Among the existing methodologies, several internationally recognized growth curves have been developed and implemented in clinical practice. One of the most widely used methods is INTERGROWTH-21st, a reference standard derived from a multinational study conducted between 2009 and 2014 in Brazil, China, India, Italy, Kenya, Oman, the United Kingdom, and the United States. These growth curves provide international standards for weight, length, and HC according to gestational age, enabling assessments of fetal growth and neonatal anthropometric measurements from the 14th week of gestation up to two years of age [57,58]. These curves have demonstrated strong utility in distinguishing the relationship between the SGA status and increased risk of hospital mortality [59].

Another widely used model is the FENTON growth chart, based on a multicenter study conducted in six high-income countries with a predominantly Caucasian population. This method determines the birth weight of preterm infants by comparing it with the weight of term newborns while adjusting for gestational age. The FENTON chart facilitates the monitoring of a child’s growth alongside the WHO growth curves [60].

The OLSEN methodology was developed from a multicenter and multiethnic study in the United States, including newborns between 22 and 42 weeks of gestational age. This method considers multiple variables, such as birth weight, length, and HC, along with estimated gestational age, sex, and ethnicity [59].

In the Latin American context, the Centro Latinoamericano de Perinatología (CLAP by its acronym in Spanish) curves serve as a standardized method for tracking the growth of newborns between 24 and 44 weeks of gestational age. These curves were developed using data from various Latin American countries, making them a region-specific reference [61,62].

Finally, the LUBCHENCO curves were the first to be developed; they were created in 1963 based on a study conducted in the United States. This method describes fetal growth between 24 and 44 weeks of gestational age [63].

Table 2 presents a comparative analysis of the aforementioned growth curves, analyzing various parameters.

## 4. Discussion

Fetal malnutrition results from a deficiency in muscle and fat accumulation during intrauterine life. It is clinically diagnosed by the presence of subcutaneous fat loss and reduced muscle mass in the newborn [34], regardless of birth weight or the weight-for-gestational age ratio. Intrauterine malnutrition triggers alterations in genetic programming, metabolism, and body composition, resulting in both short- and long-term complications. Immediate consequences include hypoglycemia, hypocalcemia, hypothermia, and polycythemia [12], while long-term issues may involve neurological disorders, learning difficulties, intellectual disability, metabolic syndrome, hypertension, obesity, and diabetes [23,24,25].

Over the years, various tools and methodologies have been used to comprehensively assess the newborn’s nutritional condition. The most commonly used include the weight-for-gestational age ratio (weight/GA) [66], length-for-gestational age ratio (length/GA), head circumference-for-gestational age PI [67], CAN score [34], BMI, MUAC, McLaren index (MUAC/HC) [32,67], and plicometry [37]. These tools offer complementary approaches that enhance the identification of fetal malnutrition and enable a more accurate classification of the neonatal nutritional status, supporting clinical decision-making.

The prevalence of fetal malnutrition varies depending on the region, country, diagnostic methodology used, and the time frame of studies. In some regions, a slight decline in prevalence has been reported since 2020 [12]. Traditionally, birth weight has been the primary diagnostic criterion for fetal malnutrition; however, this approach has limitations, as it does not assess the newborn’s body composition. Studies in Latin America estimate a fetal malnutrition prevalence of 14% to 16% when birth weight alone is used as the diagnostic criterion, whereas in developed countries, the prevalence is **below 6%.** These differences underscore the importance of implementing comprehensive assessment criteria for the accurate diagnosis of fetal malnutrition [68].

Additionally, a study conducted in Ethiopia by Sume reported a 12.3% prevalence of fetal malnutrition. This study identified that fetal malnutrition was associated with factors such as a lack of prenatal nutritional counseling, lower maternal MUAC, low maternal BMI, placental weight below 519 g, prematurity, and low birth weight [69]. Similarly, a meta-analysis incorporating data from Egypt, Nigeria, and Ethiopia found that Egypt had the highest prevalence of fetal malnutrition at 30%. In this analysis, Mussa et al. (2024) [12] reported that factors such as SGA, prematurity, and neonatal sepsis significantly contributed to this condition. Furthermore, differences in governmental nutritional intervention policies played a substantial role in the burden of fetal malnutrition across the studied countries [12]. These results highlight the geographical variability and the influence of socioeconomic determinants on the incidence of fetal malnutrition.

In 1966, the American Academy of Pediatrics coined the term “Small for Gestational Age” (SGA), marking a significant advancement in differentiating between low birth weight, prematurity, and low weight for gestational age [70,71]. Since its introduction, the diagnosis of SGA has been widely used in neonatal nutritional evaluations and assessments of its relationship with fetal or neonatal malnutrition. However, the CAN score was developed as a more objective and practical method for identifying fetal malnutrition [21].

Although, in previous decades, low birth weight was considered a key predictor of neonatal mortality, recent research has challenged this perspective. It is now known that birth weight alone is not a determining risk factor; rather, prematurity due to organ immaturity and fetal malnutrition due to metabolic alterations are the main risk factors for perinatal diseases and complications [72]. In this context, the diagnosis of fetal malnutrition has advanced beyond birth weight or even growth curves, as these criteria may be insufficient for detecting alterations in the neonatal body composition. Thus, the inclusion of the CAN score or BMI has been fundamental in improving the diagnostic accuracy for fetal malnutrition, providing a more precise assessment of the newborn’s nutritional status [35].

The CAN score has been employed in various observational studies and clinical settings to enhance the accuracy of newborn nutritional assessments [12,69,73]. In a retrospective study conducted on 77 term neonates, the CAN score was used to identify 50.3% of SGA newborns, 4.6% of AGA newborns, and 0.28% of LGA newborns who presented malnutrition, demonstrating the need to expand nutritional appraisal using this tool [39]. In the study by Mosan Raza et al., the CAN score identified malnutrition in 19.1% of neonates. A strong correlation was found between the CAN score and weight classifications (SGA, AGA, and LGA) (*p* < 0.001). Additionally, a bivariate analysis using Pearson’s test showed a significant and positive relationship between anthropometric indices (MAC/HC, BMI, and PI) and the CAN score (*p* < 0.05) [10].

In Latin America, several studies support the CAN score as the most appropriate tool for the neonatal nutritional assessment. For instance, a study conducted in Peru with a sample of 294 term newborns found that the CAN score had the best prognostic value for predicting neonatal morbidity. It was observed that newborns exposed to fetal malnutrition had a significantly higher risk of developing morbidity compared to those without fetal malnutrition (*p* < 0.05) [38].

Similarly, in Ecuador, a 2018 study involving 104 term neonates reported a malnutrition prevalence of 38%. This study used indicators such as weight-for-gestational age, length-for-gestational age, the McLaren index, and the CAN score. The results indicated that the CAN score was the most effective tool for determining the nutritional status of the newborn [74].

Various studies have compared the CAN score with other nutritional evaluation methods, such as growth curves, and have demonstrated better results in detecting neonatal malnutrition. A prospective study conducted by Alberca Garcia et al. (2019) on 93 term newborns with AGA (determined using Intergrowth 21st growth curves) found a malnutrition incidence of 16.1%. These malnourished neonates would not have been identified without the application of the CAN score [75]. This result is comparable to malnutrition prevalence rates ranging from 12% to 19%, as reported in studies from various regions worldwide [10,12,68], which used not only birth weight and growth curves but also additional nutritional assessment tools, such as the CAN score, BMI, PI, and plicometry.

Although the CAN score has proven to be an effective tool for assessing the neonatal nutritional status, some studies suggest that BMI may have greater sensitivity in detecting malnutrition. To determine the best method for the neonatal nutritional evaluation, Thomas and collaborators conducted a study on 1000 neonates born in Mangalore, India, comparing the performance of BMI and the CAN score. The results showed that 25.9% of newborns had a BMI below the 10th percentile (malnutrition) and 9.9% had a BMI below the 3rd percentile (severe malnutrition). On the other hand, when the CAN score was used for the assessment, 31.9% of neonates were diagnosed with malnutrition. The difference in test performance was statistically significant (*p* < 0.001), revealing that BMI was a better indicator of the nutritional status [76].

Similarly, Tiwari A.K., in a study involving 349 term neonates, applied various indices (BMI, PI, McLaren Index, and CAN score) to determine the most effective method for evaluating the neonatal nutritional status. It was found that BMI had the highest sensitivity (75.7%, *p* < 0.001), and when combined with the PI, the sensitivity increased to 89.1% [77]. In 2008, Brock et al., in a study conducted with 2406 neonates with AGA, used the growth curve developed by Alexander et al. as a reference to establish BMI values for each gestational age [24]. These BMI curves for newborns are now a valuable tool for determining the neonatal nutritional status.

Furthermore, Gupta conducted a study in India, comparing the effectiveness of different methods for neonatal nutritional assessments and using the CAN score as the gold standard for diagnosis. The study evaluated birth weight, weight-for-gestational age, length, HC, MUAC, PI, and the CAN score. The cut-off points for diagnosing malnutrition were: PI <2.2 g/cm^3^, McLaren Index (MUAC/HC) of 0.27, and BMI of 11.20 kg/m^2^. The results revealed that 4% of neonates with AGA were malnourished, while the prevalence reached 42.9% in neonates with a low birth weight for gestational age [78].

Furthermore, when evaluating the sensitivity and specificity of different tools for detecting neonatal malnutrition, the results vary across indicators and studies (Table 3). These findings highlight the need for a complementary assessment approach, as no single tool is sufficient. In low-resource settings, simple, low-cost methods like the CAN score are especially useful. Adapting strategies to the local context improves the detection and management of neonatal malnutrition. In terms of the sensitivity and specificity of each indicator, the McLaren Index (MUAC/HC) has a sensitivity of 39.68% and specificity of 75.6%, the CAN score has a sensitivity of 71.88% and specificity of 89.9%, BMI has a sensitivity of 84.48% and specificity of 75.5%, and the PI has a sensitivity of 53.76% and specificity of 84.14% [78]. BMI and the CAN score were the most sensitive tools for detecting neonatal malnutrition, emphasizing the importance of using multiple tools for neonatal nutritional assessments, as each indicator has strengths and limitations [78].

On the other hand, plicometry has been explored as a complementary tool in determining the nutritional status. Its relationship with other methods, such as the CAN score, has been studied to establish its role in detecting neonatal malnutrition. For example, Kafle and his collaborators conducted a neonatal skinfold study on 370 neonates and compared it with the CAN score and anthropometry. Five measurements were taken (biceps, triceps, upper iliac, quadriceps, and subscapular region), and the average values found were 4.3 ± 1.61 mm and 4.18 ± 0.72 mm in males and females, respectively. All the measured skinfolds correlated with the CAN score, but the sum of the five had the highest correlation. However, this study lacked the necessary sensitivity or specificity to replace the CAN score [40].

Similarly, another study compared plicometry with the CAN score. The CAN score was correlated with each of the five skinfold measurements taken (biceps, triceps, upper iliac, quadriceps, and subscapular region). All measurements were on the right side of the body. The sum of the values for these five skinfolds was 3.34 ± 0.96 mm and 4.21 ± 0.35 mm (*p* < 0.001) in malnourished and well-nourished neonates, respectively. This study concluded that plicometry is a valuable tool for assessing the neonatal nutritional status, but its sensitivity and specificity are lower than the CAN score [73]. 

Table 4 summarizes what has been described in the literature according to different parameters. 

These studies demonstrate that neonatal nutritional assessment cannot be performed with just one parameter but rather requires a complementary evaluation using more than one instrument or methodology to increase the accuracy of the diagnosis [35]. The implementation of more sensitive and specific methods, such as the CAN score and the neonatal BMI, allows for the more precise identification of neonates at risk. This strategy not only optimizes resource allocation in maternal–child health programs but also contributes to reducing neonatal morbidity and mortality.

## 5. Future Directions

Fetal malnutrition remains a public health challenge with a variable prevalence, depending on the socioeconomic context and the methodologies used for diagnosis. In this context, early identification and comprehensive management of fetal malnutrition are crucial to achieve the global goal of reducing mortality by 2030 [79]. Additionally, the inclusion or development of a multiethnic BMI curve applicable to different neonatal populations may provide an accessible and standardized reference to enhance nutritional evaluations across diverse settings.

Future research on neonatal nutritional assessments should focus on designing and validating standardized, multicenter diagnostic protocols that integrate both clinical observations and anthropometric measures. As technology continues to evolve, innovative tools such as portable digital imaging systems, machine learning algorithms, natural language processing, artificial neural networks, and biochemical markers offer promising avenues to improve the accuracy and timeliness of malnutrition detection—particularly in resource-constrained environments. Mobile applications integrated with electronic health records (EHRs) could further improve real-time monitoring and continuity of care.

Recent studies report that machine learning models, like Random Forest, predict an infant’s nutritional status with >98% accuracy [80]. In addition, analyses based on massive EHR data, such as those performed on platforms like IBM Watson Health Explorys, have identified critical associations between malnutrition and developmental abnormalities [81]. However, a systematic review found that over 90% of these AI models are not yet applied in clinical practice [82].

Although much of the research focuses on pediatrics, these findings are highly relevant to fetal undernutrition. The integration of AI with mobile tools, EHRs, and clinical decision support systems (DSS) could transform maternal–fetal nutrition screening.

As an innovative contribution, our team has developed a clinical chatbot (https://poe.com/Nutritional-Status , accessed on 01 May 2025) in the pilot phase that, through targeted questions based on validated tools such as the CAN score, BMI, fetal growth curves, and other key indicators, guides the clinician in identifying and assessing the fetal nutritional risk. Although this chatbot is still in the testing stage, its design aims to facilitate evidence-based decision-making and provide interactive and accessible support for clinical teams.

## 6. Conclusions

Fetal and neonatal malnutrition remain major global health challenges, especially in low-resource settings. The evidence highlights the limitations of birth weight or growth curves alone, emphasizing the need for complementary tools such as the CAN score, BMI, and PI. These provide greater sensitivity and specificity, identifying malnourished neonates who may appear to have appropriate growth by standard metrics but exhibit signs of an altered body composition.

The CAN score, validated through systematic reviews and meta-analyses, is considered the gold standard for diagnosing undernutrition. Its high accuracy, low cost, and simplicity make it ideal for routine applications. Unlike other indices, it does not need to be contrasted with reference curves, allowing broader use—especially in low- and middle-income countries where prenatal data may be limited—facilitating early detection and closer monitoring.

While PI and BMI used together improve specificity, they depend on gestational age curves, complicating their implementation in rural or low-income areas [83]. Lastly, it is essential to consider not only clinical indicators using normalized data but also national health standards and broader social determinants that may influence neonatal nutrition.

## Figures and Tables

**Figure 1 nutrients-17-01642-f001:**
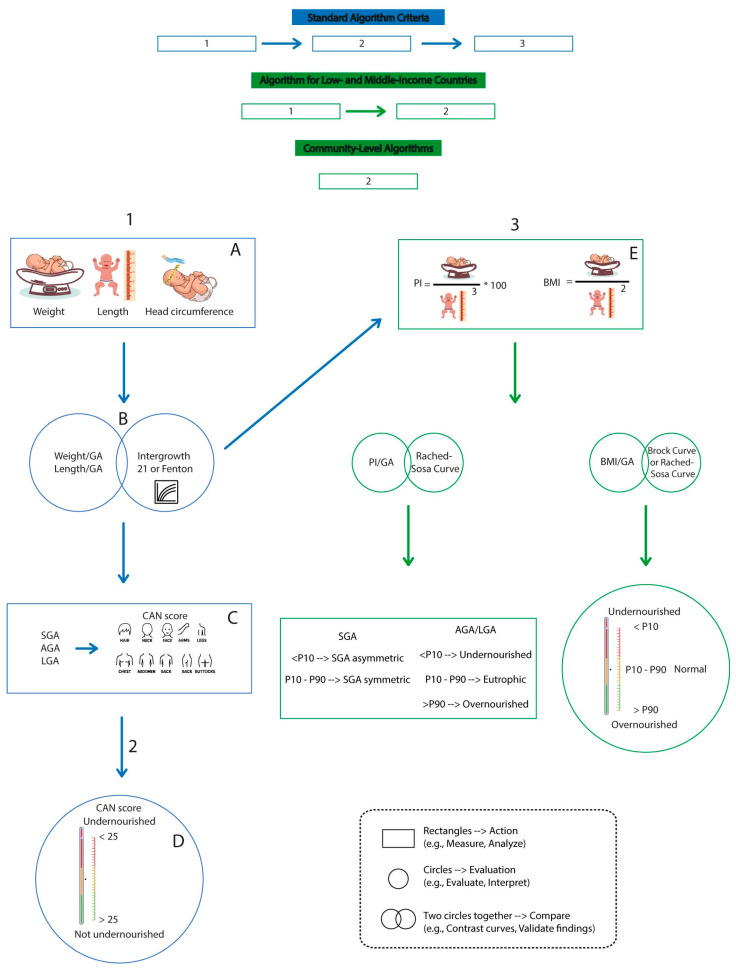
Algorithms for assessing the nutritional status using various approaches. The chart presents algorithms for assessing the neonatal nutritional status across various settings, from high-income countries to community-level contexts, using direct measurements, clinical scores (such as the CAN score), and anthropometric indices, all supported by specific reference curves.

**Table 1 nutrients-17-01642-t001:** Comparative Table of Indices Used to Diagnose the Nutritional Status of Newborns.

Tool	Type	Description	Body Composition Evaluation	Advantages	Disadvantages	Reference Standard	Cost	Required Equipment
Weight/GA	Quantitative	Compares weight to gestational age	No	Simple and enables the classification of the newborn based on growth standards.	Affected by the hydration status and requires a calibrated and accurate weighing scale.	Intergrowth 21 or Fenton	Low	Scale
Length/GA	Quantitative	Compares length to gestational age	No	Reflects lean mass, unaffected by the hydration status. and identifies children with intrauterine growth restriction (IUGR).	Less accurate due to measurement difficulty in neonates, and the use of non-standard measuring tools can compromise precision.	Intergrowth 21 o Fenton	Low	Infantometer
Head circumference/GA	Quantitative	Compares head circumference to gestational age.	No	Simple and detects cranial abnormalities.	Low sensitivity and specificity for detecting malnutrition.	Intergrowth 21 or Fenton	Low	Tape measure
PI (Ponderal Index)	Quantitative	Calculated from weight and length	Yes	High specificity in identifying malnourished newborns.	Low sensitivity	Rached-Sosa curves	Low	Scale and infantometer
CAN score (Metcoff methodology)	Qualitative	Score based on a visual observations of 9 body areas	Yes	Detects true malnutrition in AGA/LGA newborns.	An operator-dependent, subjective evaluation that may lead to inter-observer variability.	Metcoff chart (score of 9 body regions)	Low	None
BMI (body mass index)	Quantitative	Calculated from weight and length	Yes	Quick and easy to calculate.	Limited utility in preterm infants.	Brock curve, Rached-Sosa curve	Low	None
Plicometry (skin fold thickness)	Quantitative	Measures skinfold thickness	Yes	Provides a direct estimate of subcutaneous fat.	Operator-dependent, may be uncomfortable for the newborn, and requires specialized equipment.	Rodríguez curve	Medium	Caliper calibrator
MUAC (mid–upper arm circumference)	Quantitative	Determines the mid–upper arm circumference.	Yes	Highly sensitive, not affected by hydration, and useful in resource-limited areas.	Normative reference curves for neonates are limited.	Rolland-Cachera curve, Sasanow curve	Low	Tape measure
TC (thigh circumference)	Quantitative	Relates thigh circumference	Yes	Simple and non-invasive method.	Lacks validation in preterm infants, and limited standard reference data are available.	Merlob curve, Rached-Sosa curve	Low	Tape measure
McLaren Index (mid–upper arm circumference/head circumference)	Quantitative	Relates the mid–upper arm circumference to the head circumference	Yes	Effective in children over 3 months of age.	Low sensitivity for detecting malnutrition in neonates.	Rached-Sosa curve	Low	Tape measure

**Table 2 nutrients-17-01642-t002:** Comparison of growth curves [57,59,60,61,63,64,65].

Growth Chart	CLAP (2009)	FENTON (2003)	INTERGROWTH-21st (2014)	OLSEN (2010)	LUBCHENCO (1963)
**Population**	Multiethnic, international, no maternal risk	Multinational	Multiethnic, international, no maternal nutritional risk	Multiethnic	United States
**Design**	Prospective	Cross-sectional	Prospective	Cross-sectional	Cross-sectional
**Variables**	Weight, length, and HC	Weight, length, and HC	Weight, length, and HC	Weight, length, and HC	Weight, length, and HC
**Sex differentiation**	Yes	No	Yes	Yes	No
**Percentile classification**	10, 90	10, 90	10, 90	10, 90	10, 90

Adapted from (Papageorghiou et al., 2018 [57]), (Fenton & Kim, 2013 [60]), (Montealegre Pomar, 2021 [64]), (Guayasamín et al., 1976 [61]), (Figueras Aloy et al., 2024 [59]), (Díaz-Granda & Díaz-Granda, 2016 [65]), and (Lubchenco et al., 1972 [63]).

**Table 3 nutrients-17-01642-t003:** Sensitivity and specificity of each tool.

Tool	Sensitivity	Specificity	Reference
The CAN score compared to weight	71.88%	89.9%	[78]
BMI compared to the CAN score	86.21%	98.63%	[68]
	75.7%	92.8%	[77]
	84.48%	75.5%	[78]
PI compared to the CAN score	48.28%	98.9%	[68]
	89.1%	97.8%	[77]
	53.76%	84.14%	[78]
MUAC/HC compared to the CAN score	81.03%	95.05%	[68]
	42.9%	87.1%	[77]
	39.68%	75.6%	[78]

**Table 4 nutrients-17-01642-t004:** Summary of the literature review.

Study	Population	Research Type	Sample Sizes	Prevalence of Malnutrition	References
CAN score and BMI birth centiles	Pediatric unit, KS Hegde Medical Academy, India	Cross-sectional and descriptive	1000	25.9% <10th percentile; 9.9% <3rd percentile (severe malnutrition)	[76]
CAN score and various anthropometric parameters	Bacha Khan Medical Complex, Pakistan	Observational and prospective	130	40.88% fetal malnutrition	[72]
CAN score	Debre Markos Hospital, Ethiopia	Cross-sectional	414	12.32% were malnourished	[69]
CAN score	Tertiary care, North India	Prospective	250	84% were malnourished by the CAN score	[78]
CAN score	Hospitals in Ethiopia, Nigeria, and Egypt	Systematic review and meta-analysis	5356	19% with malnutrition by the CAN score	[12]
CAN score and anthropometric criteria	Ghaem Hospital, Iran	Cross-sectional and descriptive	3367	The CAN score identified fetal malnutrition in 19.1% of neonates.	[10]
CAN score and anthropometric indices	Tertiary care in Kolkata, India	Prospectiveand observational	349	20.1% with malnutrition	[77]
CAN score	Hospital de Sullana, Perú	Prospective, cross-sectional, and descriptive	93	16% with clinical malnutrition	[75]
CAN score	Hospital Cochabamba, Bolivia	Cross-sectional and observational	53	17% with malnutrition	[35]
CAN score and anthropometric parameters	Hospital de Barcelona, Spain	Retrospective	14.477	According to the CAN score, 7.6% had malnutrition.	[39]
CAN score	Hospital Regional Docente las Mercedes, Perú	Analytical prospective cohort	294	15.6% with malnutrition; 55.6% of SGA, 16.7% of AGA, and 2.2% of LGA newborns.	[38]
CAN score and anthropometric parameters	Debre Markos Hospital, Ethiopia	Cross-sectional and observational	422	13.74% with malnutrition by the CAN score	[68]

## Data Availability

No new data were created or analyzed in this study. Data sharing is not applicable to this article.

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
