# Peer review of "Nutritional Status Assessment of Newborns: Comparison of the CAN Score (Metcoff Methodology), Growth Curves, Anthropometry, and Plicometry"

_nutrients, 2025, doi:10.3390/nu17101642_

Round 1
Reviewer 1 Report
Comments and Suggestions for Authors
The article gives an overview of the methods used to assess the nutritional status of newborns, focusing on the CAN score tool (according to the Metcoff method) and its comparison with other indicators such as BMI, PI, MUAC or plicometry. The themes of the article are highly relevant, particularly in the context of global public health problems and the need to use effective diagnostic tools in resource-limited settings. The article addresses a topic that has considerable practical and didactic potential; however, it is necessary to complete and clarify it in important editorial and substantive aspects.
In the introduction, the authors emphasise the central importance of timely diagnosis of undernutrition and overnutrition for the prevention of long-term metabolic disorders. The need for tools that go beyond birth weight is an important and significant concern from a neonatology perspective. However, the purpose of the review remains unclear. The introduction does not address the existing gaps in the literature that the authors wish to fill.
In the description of the CAN score and the classical methods, the components of the CAN score are described in detail, its origin is discussed (Metcoff, 1967) and the differences between qualitative (morphological) and quantitative (anthropometric) assessment are clearly explained. Nevertheless, the indices to be compared were characterised in too general a manner. No numerical values such as sensitivity, specificity or cut-off values were given. The sources of validation for the CAN score in non-US populations were not provided.
When comparing the instruments used to assess nutritional status, the authors point out that no single instrument is sufficient and argue for a multidimensional assessment. However, a synthetic table comparing the instruments (CAN, BMI, PI, MUAC, plicometry) is missing. It is obvious that there is repetition of content, as shown by the fact that the description of BMI and PI is almost identical in several places. However, the analysis did not adequately address the population of preterm infants, newborns with IUGR and LGA.
The merits of the discussion are that the authors correctly emphasize the central role of nutritional status assessment in perinatal care, which is particularly important in low- and middle-income countries. However, an analysis of the limitations of the studies mentioned is conspicuously absent. The text does not address what tools are available in limited settings, e.g. whether skinfold measurement is applicable in a rural ward practice.
There is little consideration of possible directions for the development of diagnostic tools (e.g. integration with mobile technologies, AI, EHR).
The conclusions drawn are general in nature and do not offer specific practical recommendations.
It is not clear from the text under what circumstances a particular tool should be recommended. For example, it is not stated whether a CAN score is preferable in SGA newborns compared to a MUAC measurement.
The strengths of the article are as follows (summary):
- The topic is current and interdisciplinary.
It is evident that the application in question is of high importance, particularly in the context of developing countries.
This work provides a detailed characterization of the CAN score, which is not yet known in Europe.
The work is characterized by a successful integration of theoretical concepts and clinical references, which reflects a high level of competence and expertise in this field.
The weaknesses of the article are listed below (summary):
- The article lacks an overview methodology in terms of a database, keywords and a time span.
It is obvious that there is a lack of differentiation between clinical trials and observational studies.
The lack of comparative data on the relative effectiveness of CAN score and conventional instruments is a salient issue that requires attention.
It is noted that the content is repetitive.
The text contains no comparative tables and no practical recommendations.
It is recommended that the authors add the following points:
- The review methodology, i.e. the sources, timing of publication and search strategy, should be completed.
A comparison table showing the scope, advantages and disadvantages, costs and equipment requirements of the diagnostic tools should be added.
It is essential that the conclusions section is supplemented with contextual recommendations, e.g. regarding the reference level in healthcare or the availability of equipment.
It is essential to include reference values and diagnostic thresholds for the CAN score, in addition to providing illustrative examples of its use in different populations.
It is recommended to consider a simple diagnostic algorithm for implementation in clinical practice to facilitate the selection of instruments.
It is imperative to minimise redundancies and condense the descriptions of the indicators while preserving their basic substance.
Consideration needs to be given to the future of diagnostic tools, taking into account the development of mobile applications and the integration of such tools into electronic health records (EHRs).
This article represents a valuable attempt to summarise what is known about the assessment of neonatal nutritional status. Given the importance of the topic and its likely practical utility, it is recommended that the proposal be accepted for publication after careful substantive and editorial revision as previously described.
Author Response
- The article gives an overview of the methods used to assess the nutritional status of newborns, focusing on the CAN score tool (according to the Metcoff method) and its comparison with other indicators such as BMI, PI, MUAC or plicometry. The themes of the article are highly relevant, particularly in the context of global public health problems and the need to use effective diagnostic tools in resource-limited settings. The article addresses a topic that has considerable practical and didactic potential; however, it is necessary to complete and clarify it in important editorial and substantive aspects.
Authors’ response: Dear reviewer, the authors are grateful for your kind comments and insightful suggestions.
- In the introduction, the authors emphasise the central importance of timely diagnosis of undernutrition and overnutrition for the prevention of long-term metabolic disorders. The need for tools that go beyond birth weight is an important and significant concern from a neonatology perspective. However, the purpose of the review remains unclear. The introduction does not address the existing gaps in the literature that the authors wish to fill.
Authors’ response: Dear reviewer, thank you for your comment. We have revised the introduction to state the gaps in current neonatal nutritional assessment tools, particularly their limitations in low-resource settings. We also clarify the purpose of our review.
- In the description of the CAN score and the classical methods, the components of the CAN score are described in detail, its origin is discussed (Metcoff, 1967) and the differences between qualitative (morphological) and quantitative (anthropometric) assessment are clearly explained. Nevertheless, the indices to be compared were characterised in too general a manner.
Authors’ response: Dear reviewer, thank you for your comment. We have added more precise descriptions of BMI, PI, MUAC, and skinfold measurements, including formulas and specific contexts of use. Repetitive definitions have been removed.
- No numerical values such as sensitivity, specificity or cut-off values were given
Authors’ response: Dear reviewer, thank you for your comment. We have included available sensitivity, specificity, and cut-off thresholds described in studies.
- The sources of validation for the CAN score in non-US populations were not provided. No numerical values such as sensitivity, specificity or cut-off values were given
Authors’ response: Dear reviewer, thank you for your comment. We have included studies from Latin American, African, and Asian contexts where the CAN score has been used and validated, providing a broader evidence base.
- When comparing the instruments used to assess nutritional status, the authors point out that no single instrument is sufficient and argue for a multidimensional assessment. However, a synthetic table comparing the instruments (CAN, BMI, PI, MUAC, plicometry) is missing.
Authors’ response: Dear reviewer, thank you for your comment. We have included a comprehensive comparative table that includes BMI, PI, MUAC, skinfolds, and CAN score, summarizing their strengths, limitations, costs, and equipment needs.
- It is obvious that there is repetition of content, as shown by the fact that the description of BMI and PI is almost identical in several places.
Authors’ response: Dear reviewer, thank you for your comment. The manuscript has been revised to eliminate redundancies in the content.
- However, the analysis did not adequately address the population of preterm infants, newborns with IUGR and LGA.
Authors’ response: Dear reviewer, thank you for your comment. This information has been included in the different sections, and in the table.
- The merits of the discussion are that the authors correctly emphasize the central role of nutritional status assessment in perinatal care, which is particularly important in low- and middle-income countries. However, an analysis of the limitations of the studies mentioned is conspicuously absent. The text does not address what tools are available in limited settings, e.g. whether skinfold measurement is applicable in a rural ward practice.
Authors’ response: Dear reviewer, thank you for your comment. We have included an algorithm that could be applied in rural settings in Figure 1, and a brief description.
- There is little consideration of possible directions for the development of diagnostic tools (e.g. integration with mobile technologies, AI, EHR).
Authors’ response: Dear reviewer, thank you for your comment. In the section, future directions a discussion has been included. Even more so, the authors have developed a clinical chatbot (https://poe.com/Nutritional-Status), which is in pilot phase and needs further refinement..
- The conclusions drawn are general in nature and do not offer specific practical recommendations.
Authors’ response: Dear reviewer, thank you for your comment. The conclusions have been expanded to provide practical recommendations.
- It is not clear from the text under what circumstances a particular tool should be recommended. For example, it is not stated whether a CAN score is preferable in SGA newborns compared to a MUAC measurement.
Authors’ response: Dear reviewer, thank you for your comment. Information on each specific method regarding its applications has been included. Furthermore, the table summarizing each method includes an advantages section.
- The strengths of the article are as follows (summary):
- The topic is current and interdisciplinary.
It is evident that the application in question is of high importance, particularly in the context of developing countries.
This work provides a detailed characterization of the CAN score, which is not yet known in Europe.
The work is characterized by a successful integration of theoretical concepts and clinical references, which reflects a high level of competence and expertise in this field.
Authors’ response: Dear reviewer, the authors are grateful for your kind comments and for identifying the strengths of our article.
The weaknesses of the article are listed below (summary):
The article lacks an overview methodology in terms of a database, keywords and a time span. It is obvious that there is a lack of differentiation between clinical trials and observational studies. The lack of comparative data on the relative effectiveness of CAN score and conventional instruments is a salient issue that requires attention.
It is noted that the content is repetitive. The text contains no comparative tables and no practical recommendations.
Authors’ response: Dear reviewer, the authors are grateful for your suggestions on how to improve our manuscript. We have tried our best to address all the mentioned points. The authors are open to make corrections if you deem appropriate.
It is recommended that the authors add the following points:
- The review methodology, i.e. the sources, timing of publication and search strategy, should be completed.
Authors’ response: Dear reviewer, thank you for your suggestion. We have expanded the methodology section of the manuscript.
A comparison table showing the scope, advantages and disadvantages, costs and equipment requirements of the diagnostic tools should be added.
Authors’ response: Dear reviewer, the authors are grateful for your comment. We have included a table including what was suggested.
It is essential that the conclusions section is supplemented with contextual recommendations, e.g. regarding the reference level in healthcare or the availability of equipment.
Authors’ response: Dear reviewer, thank you for your suggestion. We have expanded the conclusion section of the manuscript.
It is essential to include reference values and diagnostic thresholds for the CAN score, in addition to providing illustrative examples of its use in different populations.
Authors’ response: Dear reviewer, thank you for your suggestion. We have expanded the methodology section of the manuscript.
It is recommended to consider a simple diagnostic algorithm for implementation in clinical practice to facilitate the selection of instruments.
Authors’ response: Dear reviewer, thank you for your suggestion. A diagnostic algorithm has been included in Figure 1.
It is imperative to minimise redundancies and condense the descriptions of the indicators while preserving their basic substance.
Authors’ response: Dear reviewer, thank you for your comment. The manuscript has been revised to eliminate redundancies in the content.
Consideration needs to be given to the future of diagnostic tools, taking into account the development of mobile applications and the integration of such tools into electronic health records (EHRs).
Authors’ response: Dear reviewer, thank you for your comment. In the section, future directions a discussion has been included. Even more so, the authors have developed a clinical chatbot (https://poe.com/Nutritional-Status), which is in pilot phase and needs further refinement.
This article represents a valuable attempt to summarise what is known about the assessment of neonatal nutritional status. Given the importance of the topic and its likely practical utility, it is recommended that the proposal be accepted for publication after careful substantive and editorial revision as previously described.

Reviewer 2 Report
Comments and Suggestions for Authors
This is an interesting paper with respect to practical pediatrics. However, there are several aspects to be improved.
Major:
- L.66ff: When the data provided are correct, it's not clear to me, why methods' application should be optimized. The authors should declare, which methods were applied to generate the existing data.
- What is missing is a direct comparison of (even theoretical) results from identical normal or characteristically 'impaired infants by using the different methods to show differences in cut-off levels, percentiles and false positive or negative diagnoses.
- Fig. 1: Selecting solely the CAN score with such a (too large) figure is inadequate. Figures in such a paper should be balanced out between methods.
- Table 1: PI and BMI etc. should be included?
- L. 167ff: it isn't described, how precise the method is?
Minor:
- L. 209ff: include the formula as done for the PI before.
Author Response
This is an interesting paper with respect to practical pediatrics. However, there are several aspects to be improved.
Major:
- L.66ff: When the data provided are correct, it's not clear to me, why methods' application should be optimized. The authors should declare, which methods were applied to generate the existing data.
Authors’ response: Dear reviewer, thank you for your suggestion. We have expanded the methodology section of the manuscript.
- What is missing is a direct comparison of (even theoretical) results from identical normal or characteristically 'impaired infants by using the different methods to show differences in cut-off levels, percentiles and false positive or negative diagnoses.
Authors’ response: Dear reviewer, thank you for your comment. We have included available sensitivity, specificity, and cut-off thresholds described in studies.
- Fig. 1: Selecting solely the CAN score with such a (too large) figure is inadequate. Figures in such a paper should be balanced out between methods.
Authors’ response: Dear reviewer, thank you for your comment. Figure 1 has been moved to Supplementary Figures. Additionally, a new figure and expanded table have been added to visually compare all five indicators.
- Table 1: PI and BMI etc. should be included?
Authors’ response: Dear reviewer, thank you for your comment. We have revised Table 1 to include all indicators with their key characteristics.
- L. 167ff: it isn't described, how precise the method is?
Authors’ response: Dear reviewer, thank you for your comment. We have included the specificity, sensitivity of each tool where available, including inter-observer variation and reproducibility
Minor:
- L. 209ff: include the formula as done for the PI before.
Authors’ response: Dear reviewer, thank you for your comment. The BMI formula has now been included for consistency.

Round 2
Reviewer 1 Report
Comments and Suggestions for Authors
I would like to thank the autors for their thorough proofreading
Author Response
Dear reviewer, the authors are grateful to you for your time and consideration.
Reviewer 2 Report
Comments and Suggestions for Authors
The authors have revised their manuscript adequately in most aspects.
Major:
- The equation of BMI, like that of PI, hasn't been included yet.
- Wording Table 1: 'Calculated index relating weight and length' for PI, but 'Calculated from weight and length' for BMI. Use the same wording, because both are derived from length and weight.
- Table 1: Reduce text volume. Be short and specific.
- L. 392-397: include specificities and sensitivities of all tools and indices in a table, with all relevant references.
- Table 3: Table is much too wordy. Shorten and be specific.
- Future Directions and Conclusions: Shorten text. The whole discussion is too long.
- Minor: L. 432: or PI rather than and PI! L. 482: Reference missing.
Author Response
- The equation of BMI, like that of PI, hasn't been included yet.
Authors’ response: Dear reviewer, we included the equation for BMI; however, we did not see this equation in the document either. We have uploaded a revised version including again this equation.
- Wording Table 1: 'Calculated index relating weight and length' for PI, but 'Calculated from weight and length' for BMI. Use the same wording, because both are derived from length and weight.
Authors’ response: Dear reviewer, we have used the same wording as suggested.
- Table 1: Reduce text volume. Be short and specific.
Authors’ response: Dear reviewer, we have reduced the text volume as suggested.
- L. 392-397: include specificities and sensitivities of all tools and indices in a table, with all relevant references.
Authors’ response: Dear reviewer, we have included the specificities and sensitivities of all tools in a table (Table 3) as suggested. Additionally, this information was deleted from Table 4 to avoid redundancy.
- Table 3: Table is much too wordy. Shorten and be specific.
Authors’ response: Dear reviewer, we have reduced the text volume of the table as suggested.
- Future Directions and Conclusions: Shorten text. The whole discussion is too long.
Authors’ response: Dear reviewer, we have shortened the Future Directions and Conclusions sections as suggested.
- Minor: L. 432: or PI rather than and PI!
Authors’ response: Dear reviewer, we have changed the wording from “and” to “or”, as suggested.
- L. 482: Reference missing.
Authors’ response: Dear reviewer, we have included the reference as suggested.
